# Atheism of the Word: Narrated Speech and the Origin of Language in Cohen, Rosenzweig and Levinas

William Large 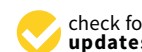

Department of Liberal and Performing Arts, University of Gloucestershire, Cheltenham GL50 4AZ, UK;
wlarge@glos.ac.uk

**Abstract:** Kant marks a fundamental break in the history of philosophy of religion and the concept of God. God is no longer interpreted as a being necessary to understand the existence of a rational universe, but as an idea that makes sense of our morality. Cohen supplements this idea with the concept of personality, which he argues is the unique contribution of Judaism. For Rosenzweig and Levinas, the monotheistic God is neither a being nor an idea, but the living reality of speech. What would the atheism be that responds to this theism? Linguistics makes a distinction between direct, indirect, and free indirect speech. In the latter form, the origin of speech is not a subject, but narrated language. It is this difference between direct and indirect speech that is missing in Rosenzweig and Levinas's description of God. It would mean that God is produced by language rather than the subject of language. What menaces the reality of God is not whether God exists, or is intelligible, but the externality of language without a subject.

**Keywords:** Cohen; Rosenzweig; Levinas; God; atheism; language

## 1. Introduction

Is there a Jewish philosophy that would be "other" than philosophy? But how can that be the case, because if it were truly other then it would not be philosophy at all, but something else? It must be both philosophical and non-philosophical. It would speak the language of philosophy, but at the same time, smuggle Jewish content into the heart of philosophy and thereby transform it from within into something other than itself. An example of such a trafficking would be Hermann Cohen's *Religion of Reason out of the Sources of Judaism*, whose title explicitly announces such an operation. In the introduction to his book, Cohen argues that a religion of reason is separate from the history of religion. History does not determine the concept of reason, rather reason determines the concept of history. If we are to understand the concept of a religion, like Judaism for example, then we cannot just rely on literary sources, because to interpret them the religion we are investigating must already have a meaning for us. We must already have a concept of Judaism even before we approach the religious texts of Judaism. This other source is reason. "*The concept of religion*," Cohen writes, "*should be discovered through the religion of reason*" (Cohen 1995, p. 5) [Emphasis in the original]. Only then does religion become a properly philosophical problem. Yet what does any religion contribute to the universality of reason except its own particularity? The answer to this question is each particularity expresses its own universality. In form at least, Cohen's strategy is the same as Kant's in *Religion within the Boundaries of Mere Reason* for Christianity, where Kant claims that Judaism, like Islam, is a mere statutory faith, and thus a particularity that only expresses the particularity of its literary sources,

Cohen will claim that Judaism has a universal significance different from Christianity. The universal, paradoxically speaking, is not a unity, but a plurality.[1]

There must be a way in which a religion can state a universal truth, but, at the same time, not lose its specificity. Each religion speaks a universal truth to humanity as its own truth. What, then, is the truth of Judaism? It is, Cohen answers, its ethics, but an ethics of a specific kind. For Kant, religion is subordinated to the universality of the moral law, which has its concrete form in the state and eventually a cosmopolitan order. Yet this sacrifices the particularity of the individual for the sake of a totality, however just that totality might be. Kant's morality only recognizes the 'I' as a member of humanity, but not as an individual. The objective form of the universal ideal of humanity is the state. What is lost in this objectification is the 'you' of the individual who stands before me. In the totality of the state, every person becomes a 'he', which is almost indistinguishable from an 'it'. What Judaism recognizes beyond Kant's morality and politics, is the individuality of an 'I' beyond abstraction. "Is it not precisely through the observation of the other man's suffering," Cohen asks, 'that the other is *changed from the He to the You*" (Cohen 1995, p. 17) [Emphasis in the original. Translation modified]. To preserve the specificity of Judaism is, then, to save the plurality of the other from the universality of the moral law and the state. The universal particularism of Judaism is the reclaiming of particularity of the individual. This is the specific contribution of the religion of Judaism to the universality of reason. It is the paradoxical expression of particularity as universal, as opposed to the universalism of the universal of Christianity, where each of us belongs to the same idea of humanity. In Judaism, each of us are singular, but universally so. The universalism of the universal is the politics of the state, whereas the universalism of the particularity of the ethics of religion.[2] Judaism is the universalism of the particular, whereas Christianity is the universalism of the universal. The universality of reason, which is now plural, requires both.

The difference between Rosenzweig and Cohen is not so much the idea of the universalism of particularity as a resistance to totality as the truth of Judaism, but the form of its presentation, and this changes the relation between religion and reason. If Cohen rejects the literary sources of Judaism for the sake of reason, then Rosenzweig does precisely the opposite. He discards reason for the sake of literary sources. The only meaning of Judaism, in its particularity, is its literary sources. Cohen can translate the particularity of Judaism into the universality of reason because he has already sacrificed the particularity of Judaism by turning it into a philosophy that can be added onto the history of philosophy. If Cohen is still writing a philosophy of religion and preserving Judaism much in the way that Kant saves Christianity, though as the universalism of particularity that preserves the individual within the universal, then Rosenzweig's *Star of Redemption* is a "cryptotheology" against philosophy, and like every "cryptotheology" what it appeals to against philosophy is a particularity that cannot be taken up by a universal that claims to be the origin of every particularity.[3] Both Cohen and Rosenzweig

---

[1] For Kant, the superiority of Christianity over Judaism is that it is a moral and not learned religion based on books. He will go as far as to say that Judaism is not a real religion at all, but merely a political constitution that expresses the will of a people (see Kant 1998, pp. 130–33). It is for this reason that Cohen, who argues that Judaism does have a universal significance, like Christianity, claims there must be a universal concept of Judaism separate from its literature. The source of Kant's anti-Semitism is the universalism of reason. If reason is defined, in advance, as the universality of the moral law, which conceals the particularity of Western philosophy, then any other culture must be found wanting.

[2] There is another politics in *Religion of Reason*, which is messianic. There is a tension in this book between the politics of the state and the politics of religion. Socialism is both the politics of the state and religion. It is a politics of the state because it is part of contemporary politics, but it is also a politics of religion because it works toward a future of justice for all, and the latter requires more than the action of a state. Messianism is an "ethical socialism" (Cohen 1995, p. 311).

[3] It is a "cryptotheology" and not a theology because it shatters the universality of philosophy from within. If Cohen attempts to universalize the particularity of Judaism, then Rosenzweig 'particularizes' the universal. Every universal is a hidden particularity, even the apparent universality of philosophy. It has no more claim to universality than any other particularity. The universal is then transformed into a patchwork of particularities in relation with one another. Cohen sees the particularity of Judaism taken up into the unity of reason, Judaism adds what philosophy lacks through its Greek heritage, whereas Rosenzweig understands the plurality of the universal as heterogenous. The universal is universal as the plurality of particularities and there is no vestige of a universality that communicates between them. There is no "meta-universality". There is no universalism of the universal. This mirrors the difference of their presentation of particularity. For Cohen,

are appealing to a particularity of Judaism, but one from the viewpoint of the universality of reason so the particularity of concept can be subsumed into a universal, whereas the other appeals to a particularity against every concept, even the idea of the unity of reason. If a religion is to claim its own particularity against the universality of reason, then it does so by appealing to the distinctiveness of its own experience. As soon as it enters the field of the universal, then it has already lost. But it also must show that that the universal of philosophy too hides its own particularity. There is no universality of reason that is not the triumph of a particularity over all other particularities. The history of reason is always the history of a tradition that conceals its own particularity by claiming its universality as its origin rather than the result of its history.

For Cohen, Judaism is a concept, however paradoxical that concept might be, but for Rosenzweig, it is an experience of a revelation, and a literary tradition that preserves the particularity of this experience. The difference between the "I" and the "you", this "relation without relation", which is the particularity of Judaism, is not conceived by reason, but is produced through the concrete living reality of speech.[4] Cohen still thinks of the other as another "I" through the concept of individuality, which would be the same for both me and the other, whereas for Rosenzweig, the other is different from me because they speak to me before I have any thought of them, which would make us equivalent. In this address, I am transformed from the indifference of the third person, where everyone is interchangeable, into a singular "you". "The I discovers itself," Rosenzweig writes, "at the moment where it affirms the existence of the you, through the question of the 'where' of the you" (Rosenzweig 2005, p. 189) [Translation modified].

There is a difference between speaking and what is spoken about. In speaking, the individuality of the speaker is revealed, as opposed to the universality of the concept. The living reality of speech is the existence of the "you" who speaks to an "I". Living speech, as opposed to the dead speech of concepts, is a dialogue. Someone is speaking to someone else. Thinking thinks thinking. It imagines itself as a monologue, but there can be no speaking without someone being spoken to. It is not that I relate to you and recognize your particularity, rather a "you" addresses me first and through that address the "I" is produced. The "I" is second, not first. This asymmetry of ethical relation in living speech is repeated in Levinas's description of the ethical relation. The other speaks. The other is present in the words they speak. Only in this way can we distinguish between ethics and ontology. Ontology belongs to the visible; ethics to speech. The difference between Levinas and Rosenzweig is that Rosenzweig begins with God and ends with the other, whereas Levinas starts with the other and ends with God. The first relation of speech for Levinas is not between God and me, but the other who addresses me, and only subsequently do I arrive at an ethical meaning of God.[5]

In Kant, God is an idea as part of the universal value of humanity. The universal value of humanity is expressed concretely in the real possibility of a just world whose existence is only possible through God's providence. In Judaism, God is not an idea, but a proper name. God is a relation to an individual

---

the particular is still a concept, for Rosenzweig it is an experience. There are vestiges of Cohen's approach in Rosenzweig, however, where some particularities are not particular enough to participate in this plural universality, which explains the belittling of Islam throughout the *Star of Redemption*. The universalism of the particular significance of both Judaism and Christianity is explained against the "false" particularism of Islam, which echoes Kant's own criticism of Judaism as merely a religion of books. "Islam," Rosenzweig writes, "is a religion of the book from the first moment" (Rosenzweig 2005, p. 180). For the significance and meaning of "cryptotheology" for Jewish philosophy (see Bielik-Robson 2014). Our aim in this essay is to show that no religion can escape a religion of books. Rosenzweig's heterogeneity is a bounded one, which only allows Judaism and Christianity, and does so by defining Judaism specifically as living speech opposed to the unitary rational universality of Christianity.

4   This does not mean that Cohen never refers to speech in the *Religion of Reason*, but it is a speech as a "rational speech" and not concrete living speech (see Cohen 1995, p. 81).

5   As Shmuel Trigano argues, Levinas is unique in Jewish philosophy in that God is revealed in the other person and not the other person in God. In Rosenzweig, God reveals Himself to me, but in Levinas, revelation occurs in the one who receives, and not the one who gives. The other is not revealed in God, but God in the other. "What, earlier, in classical Jewish-philosophy, was attributed to God or dissociated from this infinity—insofar as it is turned toward the human being—seems, in Levinas' thought, to have to do as much with man in his moral life as with God" (Trigano 2001, p. 296).

and not the manifestation of being. This relation to God as an individual is the universal meaning of Judaism. The uniqueness of God is not that of an idea, but of an individual. Since God is an individual, the relation to God is one of love rather than thought. In thought, I compare one individual to another, whereas I love a singular individual. The universality of Judaism is the universality of the particular, which is both the particularity of the individual other, and the individuality of God. In Kant, the individual is subsumed into the particular, since everyone only has a meaning as part of a totality, but for Judaism, the particular has a universal meaning as a particular. Every individual is unique, as God is. The universal is a plurality rather than a unity. The universality of Western philosophy is as much a particularity, as the particularity of Judaism. If philosophy has a universal meaning to everyone, then so too does Judaism.[6] The universal is not made of one voice, but many voices. When Western philosophy claims to be the only voice, it is because it conceals its own particularity in the universal, and at the same time represses the plurality of its own tradition. Each philosopher, when they create their own concept, retrospectively creates the unity of the history of philosophy in the image of their thought, but the history of philosophy, too, is a plurality of voices. There is no "metaphilosophy" of philosophy.

If the relation to God is love in Judaism, then the expression of that love is speech. God speaks to me as a unique individual and I respond. The universal particularity of Judaism is not the concept of the individual, as it sometime seems to be in Cohen, but the experience of the living speech. It is a "cryptotheology" of the word, but not the word as logos, but as speech. The relation to the "you" and the "I" is asymmetrical. The "I", then, is not an individual as an act of self-positing, but is produced through its relation to another who orders them. The archetypal form of this relation is the relation to the unique God who commands me in speech through His love for me. In thought, every "I" is identical to every other "I", but only in this relation of God to me is the "I" a real existing singular individual. It is not consciousness that is the origin of the singularity of the individual, but speech, and the first speech is the words of God to me.

If living speech is the Judaic "cryptotheology" at the heart of philosophy, since it reveals philosophy as a particularity rather than the universalism it claims to be, then what is the atheism that would respond to this theism? Atheism is not something that assails this "cryptotheology" from the outside, for it is not the atheism of an idea. Kant's moral theism is a reaction to the threat of nihilism occasioned by the success of the materialism of science. In an indifferent material universe, what value could humanity have? It could only be moral one, and religion expresses this moral significance through Christianity stripped of any supernatural and institutional excess. Yet this is the religion of an idea, and not the religion of Judaism. If Judaism is the correlation between the individual and God in speech, then both terms of the relation must be separate from one another if they are not to form a totality. The first word of the individual in response to God's demand is not "yes", but "no". The separation of the individual is atheism. As Levinas writes in *Totality and Infinity*, a "faith purged of myths, the monotheistic faith, itself assumes metaphysical atheism" (Levinas 1969, p. 77) [Translation modified]. Yet is metaphysical atheism the only atheism? Is there not another atheism, an atheism of the word, which neither Rosenzweig or Levinas are aware, or if so, only fleetingly, which threatens only when it is immediately warded off?

For both Rosenzweig and Levinas, the origin of language is the speaking subject, but there are two problems with this. First, this places the origin of language outside of language. The subject speaks, but the speaking subject is not an element of language. Secondly, there is a self of language

---

6 In his preface to the French translation of Mendelssohn's *Jerusalem*, Levinas stresses that the universalism of the Jewish people is its singularity. "In his universalism," Levinas writes, "he [Mendelssohn] does not forget the singularity of the Jewish people and its universal significance, which stems from that very singularity: Israel is still necessary to humanity's monotheism" (Levinas 1994b, p. 144). There is a difference between this universalism of the particularity of Judaism, and the universalism of the universal of Kant's description of Christianity. In the former, the universal is heterogenous, whereas in the latter it is homogenous.

that is not a speaking subject and that is the narrated self. If God is a proper name, rather than an idea, then God can only be a narrated self and not a speaking subject. Yet the origin of the narrated self is not a speaker exterior to language, but words, whose origin can only be other words. Both Rosenzweig and Levinas displace the priority of the "I" of representation and thought for the sake of other who addresses me in speech, but the other who speaks, speaks in the first person. The other or God addresses me as another "I", even though this "I" is not the "I" as an idea. Someone is speaking to me, but this someone is not anonymous. It is this person here speaking to me now in the present, or it is God speaking to me. Levinas describes the Other in *Totality and Infinity*, as being present in the word they speak. "Speech," Levinas writes, "consists in the Other coming to the assistance of the sign issued, attending his own manifestation in signs, remedying the equivocal by this attendance" (Levinas 1969, p. 91) [Translation modified]. What is this "attendance", but the origin of language in the speaking subject? Yet this would be to define the origin of language by that which is outside of language, for the presence of the other in speech is not the words spoken, but a revelation. Even though he defines ethics as language, Levinas will still speak of it as an "optics".[7] It is as though, at this point, with the origin of language itself, the difference between speaking and seeing no longer holds. Although speaking is not the visibility of being, the origin of language is still the visible presence of the speaker in speech, whether as the other or God, outside of the words spoken.

What matters in living speech is the presence of the speaker in their speech. Both Levinas and Rosenzweig decry the written word, for in writing, the speaking self is not present.[8] Yet there can be no God as a speaker, for all God's speech is reported speech. If God speaks to me, then God does so only through the literature of Judaism. There is no direct address of God to me in the way that the other addresses me in conversation. God speaks through the literature of Judaism, through the Talmud, or the Hebrew bible, or the liturgy of the synagogue. Nowhere, in either Rosenzweig's or Levinas's writing, does God address them directly. Indeed, you could claim that such a direct address by God would be prohibited by Judaism. Though Rosenzweig presents God's as speaking in the second part of *The Star of Redemption*, and revelation is explained as living speech, it is in fact by analogy with human speech, for God never speaks as a speaking subject, but always speaks as a narrated self. Linguistics makes a distinction between direct discourse, indirect discourse, and free indirect discourse.[9] In direct discourse, the speech of the other is directly represented from the point of view of the speaker: "He said, 'I was tired.'" In indirect discourse, the speaker is represented from the position of the narrator: "He felt tired". In free indirect discourse, however, there is no point of view, neither of the speaker nor the narrator: "He thought/said he was tired". The narrated self is the narration of speech and thought, but without a speaker. The origin of such a narrated self is not the presence of speaking subject in the words spoken, but the written words themselves. Is there not an atheism of the word, which would not be the same as the metaphysical atheism of a separated subject? For God as a proper name, the narrated God of Jewish "cryptotheology", would find its origin in these words, and not in a

---

7    "Ethics", Levinas writes, "is spiritual optics" (Levinas 1969, p. 78) [Translation modified].

8    "The unique actuality of speech," Levinas writes, "tears it from the situation in which it appears and which it seems to prolong. It brings what the written word is already deprived of: mastery. Speech, better than a simple sign, is essentially magisterial" (Levinas 1969, p. 69). As we have already noted, Judaism is differentiated from false monotheism of Islam by Rosenzweig because it belongs to writing. See footnote 3.

9    For an analysis of these linguistic distinctions and their relation to Biblical Hebrew, see (Van Wolde 1995). As Van Wolde reminds us, free indirect discourse is not possible in Biblical Hebrew, and God generally speaks in direct discourse, which is marked by the Hebrew phrase, וַיֹּאמֶר, 'and He said'. It is for this reason that Rosenzweig always represents God as speaking directly, though of course this is always reported speech through the written word. What if we were to think of language through a free indirect discourse only made possible through writing, rather than the direct speech, where free indirect discourse would merely be an anomaly? Language would then no longer be tied to the speaking subject, but to the exteriority of words. As Foucault writes of the poetry of Roussel, "Language has become circular and all-encompassing; it hastily crosses distant perimeters, but is always drawn by a dark center, never given, always elusive—a perspective extended to infinity in the hollow of words, just as the perspective of the whole poem opens both to the horizon and the very middle of the text" (Foucault 2004, p. 137) [Translation modified].

speaker, divine or otherwise, a language Blanchot describes as, "the murmur of the incessant and the interminable" (Blanchot 1982, p. 48).

## 2. Living Speech

It is not ideas that explain the origin of speech, but speech the origin of ideas. Speech is the creative origin of meaning, which is always in movement and never fixed once and for all. The reality of meaning is determined by speech, and without speech, meaning would not exist. Speech is the coming into existence of meaning through words. Words are always someone speaking to someone. Speech is always a dialogue within a given concrete social situation. Speech is always about something, and that "something" can be an idea, but speech is never just about something. It is spoken to someone. Speaking speaks to speaking. This is the genesis of every idea. Words must be captured by ideas to have a stable meaning, but stable meanings have their source in speech, which is always changing. It is because meaning has its source in speech, that the belief in fixed essences of words is a philosophical illusion.

The "I" responds to the other who speaks to them. It is the "speaking to" that is the ethical moment, and not "what is spoken about". It is I who responds to the other and not the other who responds to me. The other commands or orders me to respond. It is not I who command or order them. Such an asymmetry, where both the "I" and the other are in a relation, but not unified in a totality mediated by a third term, is only possible within of speech. In vision, individuals are reduced to attributes, which are defined in common. What is visible is not the same as speaking, even though the visible can be a theme of speaking. Echoing Rosenzweig, Levinas emphatically writes that "*the absolute experience is not disclosure but revelation*" (Levinas 1969, pp. 65–66). Revelation only occurs in speaking and not in seeing. It is first an orientation of the "I" and the other before anything is spoken about. The other addresses me and I respond. If it is a conversation, then it is not a conversation of equals. I do not address them first. The reason for this is that the "I" of speech is produced through the relation to the other. There is no "I" first in speech. This would be to confuse the "I" of speaking with the "I" of thought, as though, as Levinas puts it, in one of his Talmudic readings, there was "an inversion of the normal order" and "acting" preceded "understanding" (Levinas 1990, p. 42).

There is a truth of speech, which is not the same as the truth speech speaks about. If Levinas describes the relation of speaking first as the relation between the "I" and the other, then Rosenzweig does so through the "I" and God. Rosenzweig starts with the relation to God, and then proceeds to the social, whereas Levinas starts with the social, and then goes onto God. The form of revelation for Rosenzweig is love, but love is expressed only through speech. If creation is the visible world in which God is hidden, then in revelation, God is present as someone who speaks to me, but He speaks to me out of love for me. This does not mean that love is an attribute of God, as we speak of other attributes of God, like "All-Knowing" or "All-Powerful". To speak of God in terms of attributes is to speak of him as an object of knowledge, and not as a relation between a lover and a beloved. I do not love someone because of their attributes, rather they become beloved through this relation. "Love," Rosenzweig writes, "is not an attribute but an event", and as an event it is speaking (Rosenzweig 2005, p. 177). I know God loves me, because God speaks to me as an individual. I am addressed by God, or not at all. I hear God, or I do not. God speaks to me, or God does not. This speaking has nothing to do with whether I affirm God's existence from His essence. God is not an idea of consciousness, which I then go out looking for and find no confirmation. God first speaks, and then he exists for me, but only in that moment. It is perfectly possible that in the next moment, God will not exist, if I do not hear God's word. God "is" for me only to the extent I respond to God's word. The lover exists in the faithfulness of the beloved. As Rosenzweig cites from a passage from the Midrash, "If you testify to

me, then I shall be God, otherwise not" ([Rosenzweig 2005](), p. 185).[10] There is only God because we believe; we do not believe because there is a God.

If God only exists because I respond to God's love for me, then this "I" only truly exists in this response. Revelation is a dialogue and not a monologue. God speaks first, but I reply to God. God calls to me in His love for me, but I only come into existence by responding to this call. There is only an "I" because of a "you". The origin of the "I" who hears God's word is not the self-reflexivity of the "I think", but the "I" who responds to an address of God by answering the question, "where are you?" Only this "I" is really personal, because it is the individual who is addressed and not an abstraction. I love this person not all persons. To love all persons is meaningless, unless you love this one person before you individually. The "I" as the object of thought is only a label common to every "I" that thinks. The "I" of the "I think" is everyone and no-one. It has no name. Only in being addressed is the "I" authentically personal and individual. It is the "I" of my proper name, and not an abstract concept.

Just as I am addressed as a proper name in the question, 'who are you?', then so too does God speaks to me in his proper name. In his essay, "'The Eternal': Mendelssohn and the Name of God", which concerns the translation of the word, "God", in the Hebrew bible, Rosenzweig tells us that when God spoke to Moses, God did not speak to him as a necessary being, in the way philosophy talks about God, but through the three dimensions of direct speech: Addressing, being addressed, and as a theme (what is spoken about in speaking to someone) ([Rosenzweig 1993]()). God's direct speech, however, is reported through writing. Reported speech can only report the shock of the revelation of direct speech by distorting and doing its own violence to it. Mendelssohn's translation of God's name as "the eternal" is still too determined by his rationalism. For this reason, Rosenzweig explains it is better to translate the word "God" as 'Lord', rather than "Eternal", because "Lord" still contains the notion of an address. In Biblical Hebrew, אֲדֹנָי means 'my Lord', where the first person singular pronoun, enclitic has a sense of a vocative "offering". The revealed God of biblical monotheism is not the manifestation of a being, nor the intelligibility of an idea, but the proper name of a God who addresses me and to whom I respond.

By referring to the Hebrew expression, אֲדֹנָי, Rosenzweig is alluding to the prohibition of pronouncing God's name in Judaism, where rather than uttering the name of God as it is written in the Tetragrammaton, you say, "my Lord".[11] Levinas too explains that the meaning of this ritual is to remind Jews that the name of God is a proper and not a common name ([Levinas 1994a](), pp. 116–28). A proper name, as Kripke argues, is different from a common one, because it is not a definition of someone, but names an individual ([Kripke 1980]()). If we think of Aristotle as the man who taught Alexander, then he would still be Aristotle, even if we subsequently found out that he had not. The name sticks to an individual in an original act of baptism by a community that bestows it on them, and then it is passed down from one generation to the next, who remembers them. Proper names, precisely because they are passed down in this way, can too be forgotten and misplaced. The question for Levinas is not "does God exist?", but "why should we remember God's name?". Levinas refers to the answer given in verse Genesis 18:3, and its Talmudic reading in *Shevu'oth* 35b. אֲדֹנָי does not invoke the name of God as a being or an idea, but as the ethical obligation to the strangers who Abraham welcomes into his tent, and offers shelter and food, because his hospitality is a response both to God's word and theirs. Through the demand they make upon him, the trace of an absent God is present.[12] "The transcendence of God," Levinas writes, "is his actual effacement but this obligates us to men" ([Levinas 1994a](), p. 125).

---

10　"And if you are not witnesses, it will be as if I am not your Lord" *Pesikta de Rav-Kahana*, 12.

11　The Masoretic text of the Hebrew bible places the diacritics (*nikkud*) of אֲדֹנָי under the Tetragrammaton to remind the reader of this proscription. What is written is not what is read out aloud. This was then mistakenly transliterated as Yahweh or Jehovah, as though the Tetragrammaton were pronounced as it was written.

12　I have examined Levinas's description of the proper name of God, and its relation to Kripke's explanation of proper names as 'rigid designators', in ([Large 2013]()).

In both Levinas and Rosenzweig, narrated speech is subordinated to direct speech, yet God's word is only accessible to us through narrated speech. We do not hear God speak, but we read about those, like Moses or Abraham, who have. All testimony is through the written word. We bear witness to the proper name of God through writing (the difference between how Tetragrammaton is written and how it is pronounced can only be produced through an effect of writing). What is the relation between narrated and direct speech, and is direct speech always the origin of narrated speech, or can it be the other way around, where direct speech can only be narrated, because when we pass down the word of God, we can only do so through writing? Would such a necessity change the way we think about language, and its relation to a community? A community of speakers only survives because of the narrative of God's word. It is the written word that makes this possible and not direct speech. Does this priority of writing over speech change the way we think about language, community, and the tradition it passes down?

## 3. Narrated Speech

The proper name of God is fixed by a community of speakers who remember the original baptism of the name. If this ritual is not passed from each generation to the next through the rituals of the prohibition of pronouncing the name and its effacement, then it will be forgotten. The memory of the direct speech of God is only possible through the written word. Even when Rosenzweig describes the dialogical relation between God and humanity, as the revelation of direct speech, then God always speaks through the written word of the Hebrew Bible and the Talmud. God speaks to Moses and Moses speaks to the Israelites, but this is the reported speech of the biblical verse. God never speaks to us directly. As Stéphane Moses, reminds us, the direct speech of God in *The Star of Redemption*, is always the reported speech of the written word. God does not speak to us, except through reported speech, which is witnessed and written down.[13]

The origin of language in direct speech is the speaking subject. It is only because I speak to you that that there is speech at all, and even the other addresses me from their own subject position. Is this the case with indirect speech? In indirect speech, there are two lines of communication. There is my speech and the speech of another. I am speaking to you of what someone else has said. I say or write, "He said". Yet, in narrated speech, there is a further distinction to be made. Not only can I report what another says, but I can write in the voice of another. Linguists make a distinction between direct, indirect (or reported speech), and narrated or free indirect speech. In reported speech, I repeat what has been said by another. All direct speech has this element of reported speech, because all direct speech has a theme, which is the reported speech of another even if they are not named. Narrated speech, however, is a unique case of reported speech, whose effect can only be indicated in writing. In narrated speech, the syntactical and semantic signs of reported speech are omitted, but, more importantly, so too is the focal point of speech in a speaking subject. What is important in narrated speech is not just that I am speaking or representing the voice of another, but this voice does not have its origin in a speaking subject. The self is narrated, but there is no speaker. The narrative voice is anonymous.

It is in literature, and especially the development of the novel, that we see the most extensive use of narrated speech, and where the line between who is narrating and what is being narrated becomes increasingly blurred. In the earliest forms of the novel, the narrator speaks in the voice of the characters, and usually we jump from both voices in the same passage without any sign we are doing so. In the development of the modern novel and narrative not only is the focus on the character or narrator indistinct, but the narrator too becomes lost in the narration. Take, for example, the famous opening of Kafka's *The Trial:*

---

[13] 'Entre l'expérience personnelle de la Révélation et l'impossible saisie de *ce* qui se révèle, le texte biblique interpose son discours métaphorique comme une instance médiatrice' [Between the personal experience of Revelation and the impossible grasp of *that* which it reveals, the biblical text interposes its metaphorical discourse as a mediating instance] (Mosès 1977, p. 517) [Emphasis in the original].

Jemand mußte Josef K. verleumdet haben, denn ohne daß er etwas Böses getan hätte, wurde er eines Morgens verhaftet [Someone must have been telling lies about Josef K., for without having done anything wrong, he found himself arrested one morning]. (Kafka 1989, p. 7)

Who does this 'someone' refer to? You might think it refers to Josef K., or to an absent narrator who is telling the story and perhaps is a 'stand-in' for the real Kafka. Because we tend to think of literature as though it were like a kind of telling similar to speaking, we jump over the detail that this 'someone' has no source in a speaking subject. There is a narrated self, but these is no speaker. The shift from the first to the third person is not just a change in a point of view of a speaking subject, but the effacement of the subject as the origin of language. No one is speaking, though speaking is represented. If we think of narration as a kind of telling, then we think of language as primarily spoken, and literature as merely one possible form of spoken language, but this is to deny any difference between narrated and spoken language. In narration, language can contain sentences without a speaker, so that the question "who is speaking" becomes impossible to determine.[14]

The origin of subjectivity in narrated language is not a speaker, but the narration of a self, which cannot be traced back to any point of view. In spoken language, we can distinguish between speaking and subjectivity. If I define language as essentially speaking, then this subjectivity determines language externally since speech is an expression of subjectivity and not speech subjectivity. Subjectivity defines what it is to speak. This is the meaning of living speech. If language is essentially speaking, then speaking must be spoken by someone to someone. The expressivity of language is paramount in defining language as speech. Narrated speech demonstrates that language does not require a speaking subject. In narration, language is neither primarily spoken nor subjective because the narration of speech and thought does not require a self to embody it. It is a narrated self, made up of words, rather than a speaking subject, as the origin of the words it speaks.

To determine language by the speaking subject is to make the origin of language external to language. For the speaking subject is a point of view before it speaks. What is important is not the words spoken, but the relation of the subject to those words. I am addressed by another. It is the difference between the addresser and addressee, which expresses the essence of living speech, and not any of the words said. Moreover, this living speech is defined by the specific relation the addresser has to the words only they speak, and which the addressee lacks, which is a command or injunction. Authentic speech, as we have already seen in Levinas description of the ethical relation, and Rosenzweig's explanation of revelation, is defined by the presence of the speaker, even if that speaker is God, attending or attesting to the words they speak, rather than the words themselves. Even if a verbal expression is referred to, then it is not the words that are significant, but only the demand invoked by the presence of the speaker coming to their aid. The difference between Rosenzweig and Levinas, is that for Rosenzweig, God is a speaking subject present in his command to me. God is "I" and I am "you". God's response to Moses from the burning bush is "I will be that I will be" (Exodus 3:14). For Levinas, on the contrary, it is the other who is the speaking subject, and the proper name of God only has a meaning after this ethical relation.

Do we think language must have its origin in the speaker or words? If literature gives us an example of narrated speech, then we can generalize it to think about language as a whole. Deleuze explains that language for Foucault does not have its origin in a speaking subject, as a specific case of narrated language, but in every instance of subjectivity (Deleuze 1988, pp. 55–57). As I write this sentence to you now, I am a narrated self. The grammatical form of the speaking subject would not be the essence of language, but what language, without a speaking subject, generally makes possible.

---

14　It is important not to confuse the narrator's voice with the author's. The narration of speech and thought is not the direct address of the writer. To identify the one with the other would be to deny the difference between direct and narrated speech. What distinguishes narration is that language can contain sentences without a speaker. The "he" of narrated speech does not refer to an effaced narrator (See Banfield 1982). For the philosophical implication of this, (see Blanchot 1993, pp. 379–87). What I am here calling the "atheism of the word", Blanchot would name the "neuter".



It is not the subject of enunciation that that is the condition of the statement, but it is a variable of the statement or a given arrangement of statements. Rather than the subject being the origin of the statement, it is anonymous and impersonal *on parle* (one speaks) that speaks through the subject. The essence of language is language, and not a subject that exists outside of language and anchors it in an "I speak". It is not "I speak", which is the condition of "one speaks", but "one speaks" is the condition of the any self speaking as a self. The being of language is historically determined. There is not a universal "one speaks" that speaks through every subject, like there is one transcendental self that determines every empirical self. Rather, there is an assemblage of texts, phrases, and statements that belong to a given epoch, but this historical formation can never be traced back to a single statement by a speaking subject. The regularities of an epoch have their source in language that make possible the position of a speaking subject, but these regularities themselves emerge out of a background of an indeterminate "one speaks" or "it speaks", which in turn is historically dated. It is this impersonal or anonymous speaking that speaks through literature. Literature would not just be a special case of language, but is the being of language.

The other and God speak, but it is not what they say that is important. What is important is the presence or the revelation of the other, or God, in speech, and not the words spoken. The other or God does not babble. Narrated language reverses the relation between language and the speaker. For the narrated self has its origin in words and not a speaker, and these words have their source in a social milieu (the "there is speaking", rather than "I speak"). It is not the narrated self that is a limited case of language, and which is defined in opposition to living speech, but living speech that is the rare instance of language. For the origin of the subject that speaks is not consciousness, but inner speech, and inner speech always emerges from a social milieu. Words speak to words in a ceaseless flow of becoming, and only subsequently can language become the expression of a speaker in relation to a "you". Books, other people's words, thoughts in my head, all have the same reality. The opposition between the individual and the social is a false one, because the individual is social through and through. Speaking then speaks out of the background of an anonymity of language, where there is no conjunction of a speaker and a self, and out of which narrated selves are constructed. If Levinas and Rosenzweig say that the presence of the speaker must attend the words they speak, then we can turn this around, and reply that without the accompanying words there would be no expression. It is not the speaking subject that is first, but the narrated self. It is the social milieu, whose origin is narrated language, that makes self-expression possible, and not expression the social milieu. I speak to you, because there is speaking, words speaking to words, and not because there is an original subjective point of view who speaks, whether this point of view is the "I", the other, or God.[15]

God is not a speaking subject, but a narrated self, and if God is a proper name, then He is only so as narrated speech and thought, and not as a speaker. If God is a narrated self, then God exists only in the words others speak of him in writing. If God is no longer narrated, then he ceases to exist. God is not external to language speaking the words to another, but internal to a narrated language. In the former, God is transcendent in relation to language, whereas in the latter, God is immanent. He either speaks the words to me, as any speaking subject would, or He is spoken about. Rosenzweig writes about God as though he were an "I" speaking to a "you", but in the actual writing of *Star of Redemption* God is a "he", which is just the same as any "he" of narration.

---

[15] Heidegger was right to insist in *Being and Time* that authentic speech is a modification of "idle chatter", which is the anonymity of language that no one speaks, but he is wrong to valorize the one above the other as the being of language, for this would be to make the essence of language what lies outside of language, which would be the subject of enunciation. Heidegger ends up in the same impasse as Rosenzweig and Levinas, where language is defined by what is not language, which is the presence of a speaker in language. The essence of language then becomes silence. The authentic speaker who silently converses with themselves. It is no surprise that "idle chatter" is too associated by Heidegger with writing. "This idle talk is not confined to vocal gossip, but even spreads to what we write, where it takes the form of 'scribbling' [das 'Geschreibe']" (Heidegger 1962, p. 212).

Everything depends on whether the other or God is a speaking subject. If the speaking subject is not the origin of language, then all language is exposed to drift and decline, because there is neither a subject nor visible world to fix the meaning of words outside of language. If direct speech comes second rather than first (the "I" speaks because of inner speech, and inner speech has its origin in the narrated self of the social milieu), then what is fixed by a community of speakers, which is how Rosenzweig defines eternity as the continuity of generations, is always open to the ceaseless becoming of words.[16] There is a subtle difference between the Levinas of *Totality and Infinity*, who is largely inspired by Rosenzweig, and the later Levinas. Rosenzweig's God speaks in the first person, but the same cannot be said of Levinas after *Totality and Infinity*, where dialogue and living speech are no longer the exemplary form of alterity.[17] Levinas writes, in his essay "God and Philosophy", that the transcendence of God is a "he", rather than a "I" or a "you", since God commands me to be good through narrated language, and "is neither object nor interlocuter" (Levinas 1998, p. 69). He describes this transcendence through the neologism "illeity", which he adds is "transcendent to the point of absence, to the point of possible confusion with the agitation of the 'there is' (il y a)" (Levinas 1998, p. 69) [Emphasis in the original]. From Levinas's earliest work, the "there is" is the anonymity of being that is broken by the speech of the other who commands me. Being is neutral. Being does not speak, and rather than expressing my singularity, it overwhelms and submerges me. Why then would God be closer to the "there is" rather than the living speech? Because the proper name of God only exists in words, and the first word is spoken by no-one. We are all surrounded by words, swirling around us, and speaking through us, words endlessly responding to words, which is the "there is" of language. We are all babblers and stutterers when it comes to God. God never addresses me directly. God only has a meaning indirectly through narration, a self without subjectivity, and is always exposed to the immanent atheism of the word out of which any meaning emerges, always changing and never permanent, and forever running the risk of oblivion. Judaism lives on because of writing, but what written marks have not been forgotten? Is not this forgetting the real meaning of atheism, and not whether God exists or not, or whether we have an adequate idea of God?

**Funding:** This research received no external funding.

**Conflicts of Interest:** The author declares no conflict of interest.

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
