# Peer review of "Atheism of the Word: Narrated Speech and the Origin of Language in Cohen, Rosenzweig and Levinas"

_religions, doi:10.3390/rel9120404_

Reviewer 1 Report

This article raises a fresh problem: what is the atheism which corresponds to a post-metaphysical theology of the spoken word?  The answer given is derived from Blanchot and Foucault's account of language: free indirect discourse, as found in the narrated word.  The article presents in clear form a little-known but significant argument in philosophy of religion.

I might quarrel with the atheism described here.  In particular, the question about the essence and origin of language which frames the discussion relies on the priority of ontological questioning which Rosenzweig and Levinas call into question.  But this only advocates in favour of publication since it might provoke a substantial debate.

Pages 9 & 10 require a careful proofread prior to publication.

Reviewer 2 Report

This is an excellent essay on theisms and atheisms of language in Cohen, Rosenzweig and Levinas. It is beautifully written and manages to achieve astonishing clarity in exceptionally complex territory. At the same time, the essay is clearly immersed in both the primary and secondary literature despite wearing its exceptional scholarship and erudition lightly. For me, this essay's substantial argumentative move - to transpose the entire question of theism versus atheism onto the field of language, and more precisely, speech - is strikingly original and audacious, opening up new possibilities for reading not only its designated authors but figures such as Blanchot, Derrida and others. If I could make one suggestion for revision - which I leave entirely to the author's own discretion - it would be to perhaps make the potential links to a figure like Blanchot explicit: I detect a strong Blanchotian dimension to the atheist critique of language here, particularly in terms of his reading of the Neuter over the Levinasian Il y a, but Blanchot is very much left in the background of the essay. In summary, though, this is a  great essay and I recommend publication as it stands.